# Evaluation of Adding Natural Gum to Pectin Extracted from Ecuadorian Citrus Peels as an Eco-Friendly Corrosion Inhibitor for Carbon Steel

**DOI:** 10.3390/molecules27072111

**Published:** 2022-03-25

**Authors:** Jorge Núñez-Morales, Lorena I. Jaramillo, Patricio J. Espinoza-Montero, Vanessa E. Sánchez-Moreno

**Affiliations:** 1Departamento de Ingeniería Química, Facultad de Ingeniería Química y Agroindustria, Escuela Politécnica Nacional, Ladrón de Guevara, Quito 17-12-866, Ecuador; jorge.nunez@epn.edu.ec (J.N.-M.); vanessa.sanchez@epn.edu.ec (V.E.S.-M.); 2Escuela de Ciencias Químicas, Pontificia Universidad Católica del Ecuador, Quito 17-01-2184, Ecuador; pespinoza646@puce.edu.ec

**Keywords:** eco-friendly corrosion inhibitor, pectin, natural gums, simplex-centroid mixture design, impedance, linear polarization

## Abstract

The production and use of eco-friendly corrosion inhibitors allows valuable compounds contained in plant waste to be identified and repurposed while reducing the use of polluting synthetic substances. Pectin extracted from Tahiti limes (*Citrus latifolia*) and King mandarin (*Citrus nobilis* L.) in addition to natural gums—xanthan gum and latex from the “lechero” plant (*Euphorbia laurifolia*)—were used to create an eco-friendly corrosion inhibitor. The optimal extraction conditions for pectin were determined from different combinations of pH, temperature, and time in a 2^3^ factorial design and evaluated according to the obtained pectin yield. The highest pectin extraction yields (38.10% and 41.20% from King mandarin and lime, respectively) were reached at pH = 1, 85 °C, and 2 h. Extraction of pectic compounds was confirmed using Fourier-transform infrared spectroscopy, differential scanning calorimetry, and thermogravimetry analyses. Subsequently, a simplex-centroid mixture design was applied to determine the formulation of extracted pectin and natural gums that achieved the highest corrosion inhibitor effect (linear polarization and weight loss methods in NACE 1D-196 saline media using API-5LX52 carbon steel). Impedance spectroscopy analysis showed that the addition of xanthan gum to pectin (formulation 50% pectin–50% xanthan gum) improved the corrosion inhibitor effect from 29.20 to 78.21% at 400 ppm due to higher adsorption of inhibitory molecules on the metal surface.

## 1. Introduction

Protecting industrial facilities from corrosion involves high annual expenses; as such, extensive research has been undertaken on how to cope with this widespread problem [1,2]. The alternatives proposed over the years have included cathodic protection, the use of coatings, the reduction of metal impurity content, and the incorporation of suitable alloys; however, corrosion inhibitors are considered the most effective technique to prevent corrosion [3].

The protective action of corrosion inhibitors is very effective on metal surfaces when they are injected into the circulating feed currents; film-forming corrosion inhibitors, which decrease the corrosion rate by being adsorbed on the metal surface, are the most frequently used [4]. However, after their application, these substances become a polluting effluent in water and soil due to their chemical nature [5]. As such, relevant studies have focused on exploring eco-friendly substances with effects similar to those in conventional use, among which are polysaccharides such as pectin and tannins [6].

Pectin is a high-molecular-weight polysaccharide (methylated esters of galacturonic acid) present in the cell walls of plants and fruit peels. This biomolecule is a linear, water-soluble polymer comprising approximately 1000 monosaccharide units (degree of esterification) with a molecular weight of about 50,000–180,000 Da, [7,8]. In aqueous solutions, pectin can form gel films, which are produced by the three-dimensional bonds resulting from hydrogen bridge interactions between the hydroxyl groups of neighboring molecules [7,9]. In nature, there are varieties of fruits whose peels contain a high amount of pectin (between 4 and 30.00%) [9]; those belonging to the Citrus genera have been the most commercially employed [7].

Pectin contains functional groups (heteroatoms) that can interact with metallic surfaces through a first- or second-order bind through adsorption [10], which makes it suitable to study as a film-forming corrosion inhibitor. Several studies have examined commercial food-grade pectin’s corrosion inhibition for API-5LX60 steel and aluminum in acid media (HCl) [10,11,12]. A few studies have explored residue-extracted pectin’s corrosion-inhibiting effect; for example, Grassino et al. [13] investigated pectin extracted from tomato peel (*Solanum licopersicum*) as a corrosion inhibitor applied to tin.

In general, the corrosion-inhibiting results from pectin have been positive, but its protective performance is still lower compared to the inhibitor effect of synthetic inhibitors; therefore, options to improve its effectiveness include molecule modification or the use of several compounds in a mixture formulation [11].

Natural gums are complex substances with interesting structures and properties that promote corrosion inhibition [14]. They also have stabilizing properties that allow them to maintain a dispersed and homogeneous matrix in an aqueous medium. One of the most notable natural gums is xanthan gum, which is metabolized by *Xantomona campestris* bacteria and has been studied as a corrosion inhibitor for aluminum [14]. Other natural extracts, like latex from the “lechero” plant (*Euphorbia laurifolia*), contain terpenes and aromatic compounds [15] that could complement pectin’s corrosion-inhibiting action. Various studies have focused on this area; for example, Qiang, Zhang, Tan, and Shijin [16] demonstrated the protective effect of the complex components from Ginkgo leaf extract for X70 steel in acid media. In line with these results, the inhibitory efficiency of pectin extracted from citrus peels could be enhanced by adding natural gums (xanthan gums and latex from the lechero plant), which may result in an environmentally friendly corrosion inhibitor with higher protective performance compared to conventional synthetic products.

Thus, this study examines the use of pectin extracted from Tahiti lime (*Citrus latifolia*) and King mandarin (*Citrus nobilis* L.) peels together with xanthan gum and/or latex from the lechero plant to obtain a highly effective eco-friendly corrosion inhibitor for API-5LX52 carbon steel.

## 2. Results and Discussion

### 2.1. Evaluation of Citrus Pectin Extraction

Analysis of variance (ANOVA) and Pareto charts from pectin extraction factorial designs are shown in Table A1 (Tahiti lime) and Table A2 (King mandarin) and in Figure A1 of Appendix A. According to the Pareto charts for both types of citrus peel, an individual effect of pH and extraction temperature was observed. A hydrolyzing pH effect was evident after comparing extraction yields of each level applied; at pH = 3, approximately 3.00% of pectin was extracted, whereas at pH = 1, 41.20% was extracted from Tahiti limes and 38.10% from King mandarin in optimal conditions. These differences occurred because extraction mediums with a low pH have more hydronium ions (H_3_O^+^), which allows for a greater degree of breaking of glycosidic bonds by protopectin chains (cellulose, hemicellulose, and pectin matrix) [17]. Methacanon, Krongsin, and Gamonpilas [18] highlighted the solubilizing effect of hydrolyzing media (low pH) to obtain smaller polysaccharide molecules.

Temperature had a significant effect both individually and combined with pH. At high temperatures, molecular movement was improved, and better interaction between H_3_O^+^ and matrix molecules occurred; however, temperatures higher than 85 °C promoted pectin destruction at a pH = 1 and an extraction time longer than 2 h. With this combination, glycosidic bonds as well as ester bonds are broken, which form the base of the pectin structure [19].

Liew et al. [19] found that protopectin units require time for their structure to soften and allow hydrolysis, which was consistent with this study’s results: higher extraction yields were achieved with a longer application time (120 min). However, in the case of Tahiti lime, the temperature was not significant (Figure A1a) because after 60 min, despite being hydrolyzed, its structure did not allow more pectic compounds to be extracted (pectin yield was only 5–7% higher for 120 min compared to 60 min).

Response surface plots (Figure 1a and Figure 2b) show the pectin yield obtained by varying pH and temperature and using an extraction time of 120 min. The highest pectin yields (41.20% from Tahiti lime and 38.10% from King mandarin) resulted from pH = 1 and 85 °C. To verify the product was derived from pectic molecules, the results were complemented with a physicochemical characterization.

### 2.2. Physicochemical and Structural Characterization of Pectin

#### 2.2.1. Physicochemical Evaluation


Equivalent weight (PEq)


PEq refers to the number of galacturonic acid units (pectic chains with non-esterified carboxyl terminals) per pectin molecule [20].

The PEq of pectin according to the pH of the extraction medium is shown in Table 1. The lowest values of PEq corresponded to polysaccharides extracted at pH = 1 (488.23 ± 4.12 and 489.11 ± 5.99 for Tahiti lime and King mandarin, respectively), meaning they have shorter chains with a greater number of methyl ester groups per molecule [10]. Shorter pectin molecules have a higher probability of interaction between their heteroatoms and the active sites from the metallic surface to be protected [21]. Additionally, ester groups have a better interaction with the metallic surface than carboxylic groups because of the perpendicular orientation of the molecule as it approaches the metal [22]; this molecular behavior allows pectin chains to interact with each other to form the hydrocolloid layer to protect the carbon steel surface.


Methoxyl content (%Me) and esterification degree (%DE).


%Me represents the amount of esterified methoxyl groups from available carboxyl groups in the pectic chain [23], which ranged from 8.00 to 9.00% in all cases (Table 1). This percentage depends directly on pectin extraction method. According to Iglesias and Lozano [24] and Altaf et al. [20], using an acid hydrolysis extraction process, the resulting products have a methoxyl content between 8.00 and 12.00% and thus are classified as high-methoxyl pectin. In studies by Grassino et al. [13], Umoren et al. [10], and Fiori-Bimbi et al. [11], high-methoxyl pectin was used because of its jellifying properties and sensitivity in the presence of polyvalent cations. Regarding esterification degree (%DE), all products extracted had a %DE greater than 50.00% and so could all be characterized as high-methoxyl pectin.


Galacturonic acid (%AUA) and ash content.


Pectins utilized as corrosion inhibitors are not required to be as pure as those destined for food purposes; however, the pectin in this study had characteristics and physicochemical specifications similar to food-grade pectin [25]. According to the Food and Agriculture Organization [25], food-grade pectin should have a galacturonic acid value greater than 60.00% and a maximum of 1.00% of ash content. Pectin obtained in this study (pH = 1) had a %AUA of 79.48 ± 0.44 and 81.17 ± 0.65% (Table 1), and an ash content between 1.50 and 2.00% for King mandarin and Tahiti lime, respectively.

The results of physicochemical characterization showed that all the products obtained in the experimental design are pectic compounds, of which the polysaccharides extracted at a pH = 1 show the most favorable physicochemical characteristics (PEq and %Me) regarding their application as a corrosion inhibitor and higher extraction yield. Therefore, the best pectin extraction conditions for both types of citrus peel are pH = 1, 85 °C, and a hydrolysis time of 2 h.

#### 2.2.2. FTIR, DSC, and TGA Structural Characterization

The Fourier-transform infrared spectroscopy (FTIR), differential scanning calorimetry (DSC), and thermogravimetry (TGA) spectra of pectin extracted from Tahiti lime (TL), King mandarin (MK, and commercial pectin (Cm) are shown in Figure 2a,b. In Figure 2a, bands observed at 3330 and 2913 cm^−1^ were assigned to O-H and C-H stretching. Absorption bands of 1732 and 1643 cm^−1^ are evidenced and belong to the pectin fingerprint since they are attributable to the stretching vibration of esterified C=O and free carboxylic groups (COO=), respectively [26]. These functional groups are considered terminals of interest to corrosion inhibition in studies by Umoren et al. [10], Sato et al. [26], Urías-Orona et al. [27], and Garside and Wyeth [28], given the presence of atoms with lone pairs that would act as interaction sites with the metallic surface to be protected.

The DSC thermic profile of Cm, TL and MK pectin (Figure 2b) shows an endothermic peak between 120 and 128 °C that represents sample dehydration. The Cm profile shows two steep endothermic peaks at 186 and 208 °C, which are related to the functional groups’ bond-breaking process and result in polysaccharide chain depolymerization; this is also evident in its TGA profile, where a weight loss of 45% between 200 and 250 °C is observed [29].

Conversely, the DSC profile of extracted pectin (TL and MK) does not show steep peaks between 180 and 210 °C, and weight loss according to the TGA was 30% until 250 °C; therefore, these samples present higher thermic resistance than Cm starting at 225 °C.

### 2.3. Effect of Adding Natural Gums to Extracted Pectin on Corrosion Inhibition for Carbon Steel in Saline Media

#### 2.3.1. Determining the Optimal Formulation of Pectin and Natural Gum for Carbon Steel Corrosion Inhibition

Figure 3 presents the corrosion inhibitory efficiency for API-5LX52 carbon steel determined by linear polarization (LPR) analysis of each formulation of MK- and TL-extracted pectin (Pex) and gum (xanthan gum (Gx) and latex from the lechero plant (Lc)). Formulation F1, corresponding to the individual effect of Pex, resulted in an efficiency of 21.00%, which is low considering a concentration of 500 ppm and compared to other studies that have reported a corrosion inhibition efficiency of 50 and 60% [10,13]. This result is attributed to the non-uniform dispersion of molecules in the saline media. In contrast, binary formulations of Pex-Gx and Pex-Lc resulted in high corrosion protection efficiency.

The F8 formulation of Pex 50% and Gx 50% resulted in 78.10% of metallic protection, the highest percentage of corrosion inhibition among the mixtures. According to Ptaszek et al. [30], Gx and Pex are compatible polysaccharides whose molecular structure and electrically charged functional groups allow them to form complex homogeneous systems. Therefore, adding Gx to the formulation improved the molecular polydispersity in the saline media as well as the heteroatom interaction with the metallic surface’s active sites. Further, Gx and pectin have heteroatoms capable of interacting with the carbon steel surface while simultaneously reducing ion diffusion from the corrosive media to the metallic surface [14], particularly the carboxyl groups (Figure 4).

When comparing the FTIR spectra of the Pex 50%–Gx 50% to the effect of Pex and Gx used individually, an intense band can be seen at 1610 cm^−1^, which is related to the presence of carboxylic groups from Gx that, together with those present in Pex and an adequate polydispersity in the saline media, improve pectin’s individual performance.

Lc application in the F9 formulation (50:50 ratio with Pex) achieved high efficiency. This mixture protected 51.10% of the metallic surface, but its performance did not exceed that of Pex 50%–Gx 50%.

The incompatibility of Gx with Lc was evident in ternary formulations F2, F3, and F4, particularly in F2, in which Lc was added in a greater proportion to Gx (33% Lc and 8% Gx). Further, the addition of Lc in F2 affected the individual performance of Pex, which decreased from 21.00 to 2.10% due to the immiscibility of the mixture in the media. In the other formulations, there was no immiscibility, but the highest efficiency obtained was 39.15% with Pex 8%, Lc 8%, and Gx 8%. As both Gx and Lc are natural gums, compatibility was expected; however, it is possible that the vegetal nature of Lc and microbial properties inhibited interaction in the saline media.

##### 2.3.1.1. Open Circuit Potential (EoC) Analysis from LPR Assays

Figure 5a shows the EoC of the system with no corrosion inhibitor (blank) and with different formulations. These curves show the time necessary for the system to reach a stationary state before LPR analysis. Comparing the blank curve with the other curves, the system with no corrosion inhibitor took the shortest time to stabilize (800 s), while the mixture of Pex 50%–Gx 50% took the longest to reach a stationary state.

Calderón, Rossa, and Esteves [31] found that a shorter stabilization time with an EoC trend that moves toward positive potential values after reaching equilibrium is a sign of the formation of more stable corrosion products. Therefore, in the current study, the curves for the blank and Pex 100% show that corrosion products formed in the metallic surface, whereas the Pex 50%–Gx 50% formulation, which had the highest efficiency (78.10%), required more time to stabilize because of the stronger interaction of the inhibitor molecules with the metal surface [10]. In addition, the trend toward negative potential values of the formulation reflects a control in the system’s cathodic reaction.

##### 2.3.1.2. Linear Polarization Resistance Analysis

Figure 5b shows the linear polarization curves of the system with no inhibitor (blank) and with the proposed formulations. Among the trends, differences are visible that are caused by the variation of current intensity (I) resulting from the potential variation in the working electrode (Ewe). In the case of the blank, current intensity varied in a higher range (0.035 to 0.00 mA) than that of the system with the Pex 50%–Gx 50% formulation (0.0025 a 0.000 mA), which results from the low polarization resistance of the blank compared to the Pex 50%–Gx 50% formulation.

According to ASTM G59-97 [32], higher polarization resistance means the attenuation of the corrosion reaction; as such, it can be verified that the Pex 50%–Gx 50% formulation protects carbon steel immersed in saline media at the highest level compared to the other analyzed inhibitors.

##### 2.3.1.3. Tafel Analysis

In the Tafel plot (Figure 5c), two different patterns are evident. Regarding the cathodic reaction, there is little difference between the blank and the other systems, which is consistent with Umoren et al.’s [10] findings that the inhibitor did not affect the hydrogen reduction reaction. On the other hand, for the anodic reaction, a progressive modification between the blank (convex trend) and the other formulations can be seen; the curve for the system with the Pex 50%–Gx 50% inhibitor differs from the blank the most. This change in trend for the anodic process of the Tafel plot is a sign of anode modification for the simple deposition of the inhibitor on the metallic surface as a protective layer, which attenuates the cathodic reaction.

The corrosion potential (Ecorr) displacement in negative and positive directions is notable. In the case of Pex 50%–Gx 50%, there was a displacement of 80 mV (less than 85 mV), which is why this formulation can be considered a mixed inhibitor with a stronger effect in the anodic reaction modification (the displacement moved in the positive direction from the Ecorr of the blank). Similar results were obtained by Sangeetha et al. [33], Umoren et al. [10], Arukalam et al. [14], and Da Rocha et al. [34], who examined corrosion inhibition with banana extract, commercial pectin, xanthan gum, and citric extracts, respectively.

##### 2.3.1.4. Comparison of the Inhibitory Effect of Pex 50%–Gx 50%, Pex, and Cm

The corrosion inhibitory effect of pectin, Cm, Pex, and the Pex 50%–Gx 50% formulation (best efficiency) is presented in Table 2 and compared based on the variation of inhibitor concentration. In the case of Cm, the concentration with the highest efficiency (83.31%) is 300 ppm; as concentration increases, the inhibitory effect decreases to 52.40%. Regarding Pex, its inhibitory effect decreases starting at 200 ppm from 57.16 to 26.25%. In the case of Pex 50%–Gx 50%, the highest efficiency was 78.21% at 400 ppm. Comparing these results, the addition of Gx improved the individual inhibitory effect of Pex, as a result similar to that of Cm was achieved using just half of the amount of Pex; thus, its positive inhibitory effect for API-5LX52 carbon steel corrosion can be confirmed.

#### 2.3.2. Electrochemical Impedance Spectroscopy (EIS) Test

Nyquist curves are presented in Figure 6; the semicircles indicate the corrosion reaction response of the metal-saline media system according to frequency variation. Semicircle diameter differs, wherein the shortest represents the response in the system with no corrosion inhibitor; the semicircles lengthen as the Pex 50%–Gx 50% concentration in the saline media increases. According to Sangeetha et al. [33], the increase of Nyquist curve diameter results from a higher resistance of transference charge (Rct) (Table 3), which attenuates the electronic transference in the system through modification of the saline media-metal interphase. This interphase system modification is attributed to a double layer capacitance (C_dl_) (Table 3), which is produced by higher inhibitor adsorption on the carbon steel surface. Similar results were found by Umoren et al. [10].

The Nyquist curves do not have a perfect form since they represent actual impedance systems, so the parameter n (Table 3) allows the curves to be adjusted for analysis.

There is no official explanation for Nyquist curve deviation, but some authors, such as Umoren et al. [10], attribute this to metallic surface heterogeneity from adsorption of the corrosion inhibitor to the metallic surface and environmental interference during the analysis.

The highest value of corrosion inhibition was obtained by adding the 400 ppm formulation, resulting in an efficiency of 73.15% (Table 3). Higher inhibitor concentrations did not result in higher corrosion protection efficiency due to the absence of available active sites in the metallic surface. The results from EIS and LPR differ slightly (1 to 5%) even though both follow the same trend, with the best performance at 400 ppm.

#### 2.3.3. Weight Loss Method

According to the weight loss method results, corrosion inhibition efficiency increased with the Pex 50%–Gx 50% formulation, which is also seen in the electrochemical test results. Additionally, this method confirmed that the highest inhibitory efficiency was obtained at 400 ppm. The electrochemical tests also showed that after 400 ppm, the inhibitory effect decreased due to the lack of active sites on the metallic surface, which could be attributed to the molecule size of the polysaccharides in the formulation.

The corrosion inhibition efficiency results from the weight loss tests differ by a higher percentage than those from the LPR and EIS electrochemical results (Table 3) because the basis of the analysis of the weight loss tests is the amount of metal that has been corroded, while the electrochemical tests are based on electrical theories that associate the corrosion reaction to variations in current intensity and charge transfer resistance. However, such differences are not so great as to exclude any of the three methods used.

The LPR method with Tafel adjustment is the most widely used because it is more practical, requires less execution time, and is not highly affected by interference from outside the system. The impedance spectroscopy method allows the presence of important adsorption phenomena to be verified for the analysis of the corrosion inhibition process, which complements the LPR method by broadening the field of analysis. The weight-loss method, which for a long time was the main method to study corrosion, allows specific results of the oxidation rate of the metal to be obtained, which is very important. The disadvantage of this method is that, for field analysis, it requires months of development to obtain effective results, but when used in the laboratory, it complements the electrochemical analysis of corrosion [4].

## 3. Materials and Methods

### 3.1. Raw Material and Reagents

Pectin was extracted from peels from Tahiti lime and King mandarin purchased in Yaruquí in the province of Pichincha, Ecuador. Oxalic acid (C_2_H_2_O_4_) (Merck, purity ≥ 99%) and ethanol (C_2_H_5_OH) (Laquin. Cía. Ltd.a., purity 96% *v/v*) were used as pectin extraction and clotting media, respectively. Sodium hydroxide (NaOH) (Merck, purity ≥ 99%) and hydrochloric acid (HCl) (J.T. Baker, purity 10% *v/v*) were used in the pectin characterization, with commercial pectin (Merck, purity ≥ 99.7%) as a standard.

Food-grade xanthan gum (Laquin. Cía. Ltd.a., purity 99%) and latex extracted from the lechero plant were added to the mixture formulation with citrus pectin. Sodium chloride (NaCl) (J.T. Baker, purity ≥ 99.7%), calcium chloride (CaCl_2_), and magnesium chloride (MgCl_2_·7 H_2_O) (BDH Chemicals, purity ≥ 99%) were used for electrochemical tests.

API-5LX52 carbon steel was used to measure corrosion analyses; the composition of the carbon steel is shown in Table A3 (Appendix B).

### 3.2. Citrus Pectin Extraction

A 2^3^ factorial design with center points was carried out for the pectin acid hydrolysis with different temperatures (75, 80, 85 °C), extraction pH (1, 2, 3), and hydrolysis time (60, 90, 120 min). The response variable was pectin extraction yield (g of pectin/g of dry citrus peel used) [14,19]. Citrus peels were chopped to obtain pieces of approximately 1 cm and were dried in a forced convection oven (Memmert Universal Oven, 100, Germany) at 60 °C for 12 h (conditions determined by moisture assays). The dried citrus peels were then ground in a roller mill so they could pass through a 40-mesh sieve.

An acid hydrolysis process (reaction mechanism shown in Figure 7a) was carried out by adding 10 g of milled citrus peel to 250 mL of oxalic acid in concentrations of 5.00 g/L (pH 1), 0.25 g/L (pH 2), and 0.10 g/L (pH 3) according to the experimental design. The extraction system was subjected to temperatures and times as per the experimental design, in a reactor stirred at 500 rpm (Figure 7b). The temperature was adjusted by a vegetable oil hot bath. The hydrolyzed product was filtered manually through a cotton cloth. Ethanol 96% (*v/v*) was added in a 2:1 ratio to the hydrolyzed product so the pectin would clot and precipitate. After 24 h of precipitation, the coagulated pectin was separated with a nylon sieve and dried at 60 °C for 12 h. Finally, the extracted dried pectin was ground to pass through a 40-mesh sieve.

### 3.3. Evaluation of Physicochemical and Structural Properties

Physicochemical properties were evaluated by the acid-base titration method applied by Azad [35].

#### 3.3.1. Equivalent Weight

First, 0.5 g of pectin was moistened with 5 mL of ethanol (96% *v/v*) for 1 h in a 250 mL conical flask. Then, 100 mL of NaCl (1% *w/v*) was added, and the mixture was homogenized. This solution was titrated against NaOH 0.1 M (3 drops of phenol red as an indicator) to create the neutralized solution, S1. Equivalent weight (PEq) was determined according to Equation (1):(1)$$\mathrm{PEq}=\left(\frac{w\times 1000}{z\times \mathrm{Normality}\;\mathrm{of}\;\mathrm{alkali}}\right)\left[\;\frac{g}{\mathrm{eq}}\right],$$
where z is the volume of NaOH (mL) used to titrate S1, and w is the weight of the sample (mg).

#### 3.3.2. Methoxyl Content

First, 25 mL of NaOH 0.25 N was added to S1, and this solution was kept at 25 °C for 30 min. Then, 25 mL of HCl 0.25 N was added, and the resulting mixture (S2) was titrated against NaOH 0.1 M. Methoxyl content (%Me) was calculated with Equation (2):(2)$$\%\mathrm{Me}=\frac{k\times \mathrm{Normality}\;\mathrm{of}\;\mathrm{alkali}\times 3.1}{w},$$
where k is the volume of NaOH (mL) used to titrate S2.

#### 3.3.3. Galacturonic Acid Content

Galacturonic acid content (%AUA) was determined by Equation (3):(3)$$\%\mathrm{AUA}=\frac{176\times 0.1\left(k+\;z\right)\times 100}{w\;\times 1\;000},$$

#### 3.3.4. Esterification Degree Determination

Esterification degree (%DE) was determined by Equation (4):(4)$$\%\mathrm{DE}=\frac{176\times \%\;\mathrm{Me}}{31\times \%\;\mathrm{AUA}}\times 100\%,$$

#### 3.3.5. Ash Content

Ash content was determined using the ASTM E1755-01 method “Standard Test Method for Ash in Biomass” [36]; 0.5 g of pectin was burned in a muffle furnace at 500 °C for 20 min. The burned sample was cooled in a desiccator and weighed after 3 h.

### 3.4. Structural Analysis

Pectin extracted at optimal conditions was characterized using FTIR, DSC, and TGA. FTIR analysis was carried out using a Spectrum One FT-IR spectrometer (Perkin-Elmer, Beaconsfield, UK), with a range of 4000–400 cm^−1^ [10]; absorption bands were compared against the commercial pectin spectrum. DSC analysis was conducted between 20 and 300 °C using a nitrogen atmosphere [37]. In this case, endothermic and exothermic picks were compared between the extracted and commercial pectin. TGA was carried out between 20 and 600 °C with a temperature gradient of 10 °C/min [37]. For DSC and TGA analyses, a Shimadzu TGA-50 thermal analyzer was used.

### 3.5. Evaluation of Natural Gums Added to Extracted Pectin for Carbon Steel Corrosion Inhibition

A simple-centroid three-component mixture design was applied to extracted pectin and added gums (xanthan and latex from the lechero plant) to obtain the formulation with the highest corrosion inhibitory effect. The proposed mixtures were added to the corrosion cell at 500 ppm. The minimum percentage of pectin in each formulation was 50%, and the response variable was the inhibition efficiency obtained by the LPR technique using a BIOLOGIC SP-300 potentiostat/galvanostat, according to ASTM G-59 “Standard Test Method for Conducting Potentiodynamic Polarization Resistance Measurements” [32].

#### 3.5.1. LPR Electrochemical Evaluation

A three-electrode corrosion cell was created by adding 50 mL of saline solution (corrosion medium), prepared according to NACE 1D-196 “Laboratory Test Methods for Evaluating Oil-field Corrosion Inhibitors” [38], to a glass cell. API-5LX52 carbon steel was used as the working electrode with an estimated exposed area of 0.101 mm^2^; a graphite bar and an Ag/AgCl_2_ electrode were selected as counter and reference electrodes, respectively. The corrosion cell was bubbled with N_2_ for 5 min before each test.

The carbon steel’s oxidation potential range was determined by cyclic voltammetry (VC), which showed ±110 mV around the steady-state EoC. The corrosion system was stabilized for 20 min, and LPR analysis was performed immediately for each formulation with a potential rate of 1 mV/s. Tafel plots were drawn from test results, and the inhibition efficiency (η%) was calculated according to Equation (5):(5)$$\eta \%=\left(1-\frac{I_{\mathrm{corr}}}{I_{\mathrm{corr}}^{o}}\right)\times 100\%,$$
where $I_{\mathrm{corr}}$ is the current density in the presence of the inhibitor (mA/cm^2^), and $I_{\mathrm{corr}}^{o}$ is the current density without the inhibitor (mA/cm^2^).

To obtain the highest inhibitor concentration of the optimal formulation of pectin and gums, the mixture was analyzed using LPR in a range between 100 and 1000 ppm with intervals of 100 ppm. The same process was performed for extracted and commercial pectin, as well.

#### 3.5.2. EIS Analysis

EIS was carried out using the formulation of pectin and gums with the highest efficiency; the measurements were executed between a frequency range of 10,000.0–0.1 Hz with an amplitude of 10 mV to minimize interference [34]. The formulation was analyzed for five concentrations (values that resulted in the highest efficiency according to LPR). Using the results, Bode and Nyquist plots and corrosion inhibitory efficiency (η%) were determined by Equation (6):(6)$$\eta \%=\frac{R_{\mathrm{ct}}-R_{\mathrm{ct}}^{o}}{R_{\mathrm{ct}}}\times 100\%,$$
where R_ct_ is the charge transfer resistance (Ω/cm^2^), and $R_{\mathrm{ct}}^{o}$ represents the charge transfer resistance without the inhibitor (Ω/cm^2^).

#### 3.5.3. Weight Loss Method

Weight loss analysis was performed according to Da Rocha et al. [34] based on ASTM G31-72 “Standard Practice for Laboratory Immersion Corrosion Testing of Metals” [39]. Carbon steel samples were obtained from 50 × 25 × 20 mm API-5LX52 pipe; samples were cleaned (with ethanol and distilled water) and dried for 6 h at 30 °C. Corrosion coupons were weighed and immersed in the saline media NACE 1D-196 [38] inside a closed system. The analysis was evaluated in concentrations between 0.1 and 1.0 g/L. Carbon steel coupons were immersed for 48 h, after which they were smoothed with #50 sandpaper and washed with a rust-removing solution (NaOH 10%). Finally, samples were dried for 1 h at 30 °C and weighed. Corrosion rate (mmpy) was calculated with Equation (7) [39]:(7)$$\mathrm{Vc}=\frac{K\times W}{a\times t\times d},$$
where V_c_ is corrosion rate (mmpy), K is a constant (8.76 × 10^4^), W is weight loss (g), d is density (g/cm^3^), a is coupon exposed area (cm^2^), and t is time (h).

## 4. Conclusions

The optimal pectin extraction conditions for the selected fruits were pH = 1, 85 °C, and hydrolysis time of 120 min; yields of 38.10% and 41.20% (with respect to the dry peel sample mass) were obtained for King mandarin and Tahiti lime, respectively. Pectin with a high degree of methoxy was obtained, with an esterification percentage of 54% and 55% for King mandarin and Tahiti lime, respectively.

The addition of xanthan gum to the extracted pectin improved its individual inhibitory effect, from a protection level of 26% to 78%, evaluated at a concentration of 500 ppm at room temperature. The concentration of the Pex 50%–Gx 50% formulation that produced the strongest corrosion-inhibiting effect was 400 ppm with a metal protection percentage of 78%.

Future studies should evaluate the corrosion-inhibiting effect produced by pectin’s antioxidant capacity. In addition, it is important to continue the search for a formulation with a more effective inhibition effect compared to that obtained in this study, from pectin modified with compounds such as amines and carboxymethylcellulose, to achieve higher corrosion protection.

## Figures and Tables

**Figure 1 molecules-27-02111-f001:**
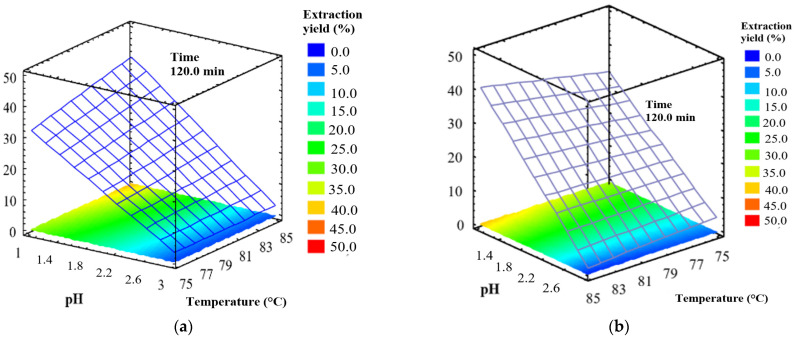
Response surface plots of pectin yield as a function of pH and temperature: (**a**) Tahiti lime (**b**) King mandarin.

**Figure 2 molecules-27-02111-f002:**
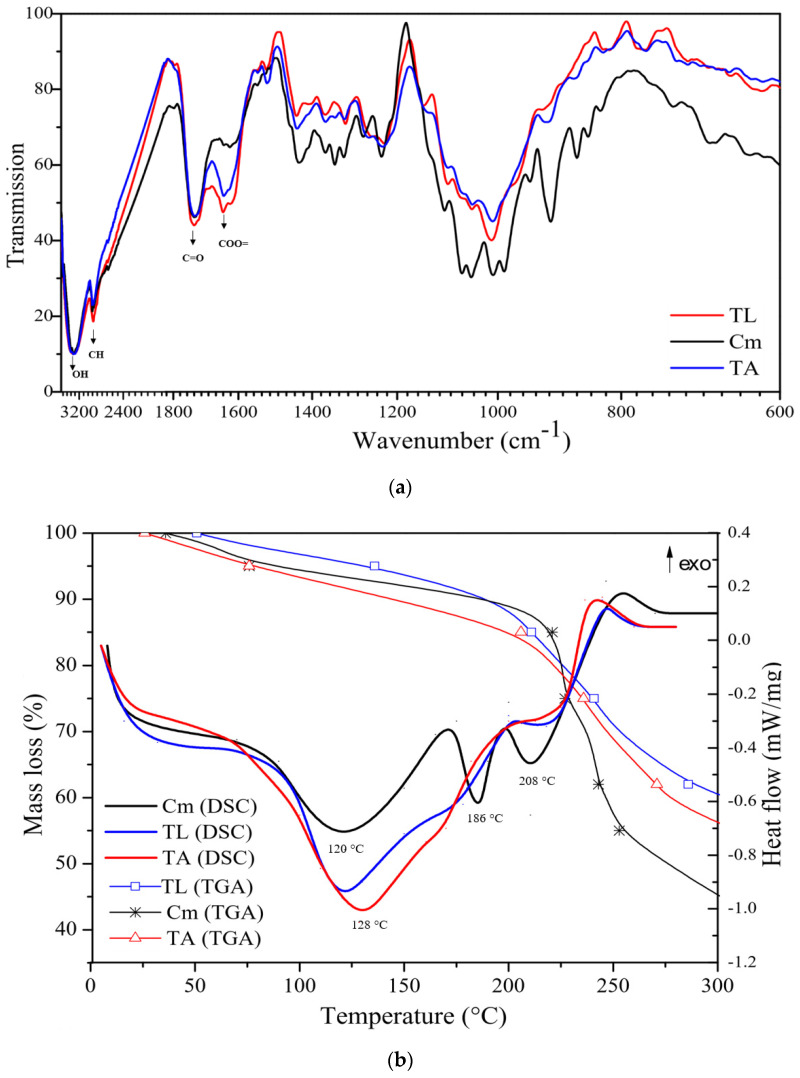
Tahiti lime (TL), King mandarin (MK), and commercial pectin (Cm): (**a**) Infrared spectra; (**b**) DSC and TGA thermal analyses.

**Figure 3 molecules-27-02111-f003:**
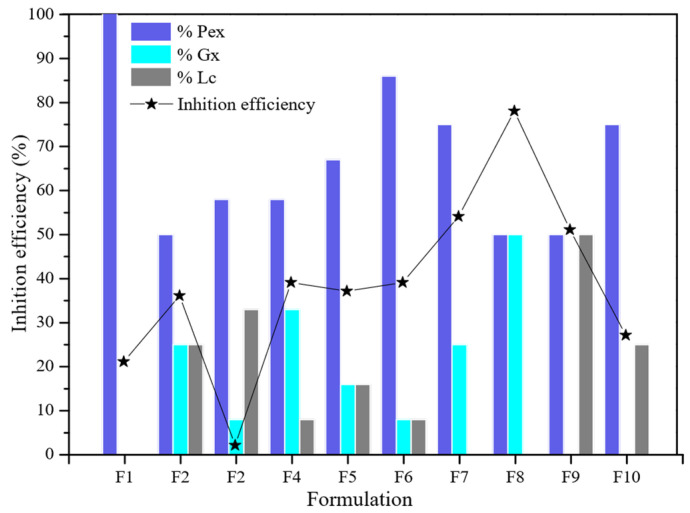
Corrosion inhibition efficiency graph for each of the extracted pectin (Pex), xanthan gum (Gx), and lechero plant (Lc) mixture variants in the experimental design.

**Figure 4 molecules-27-02111-f004:**
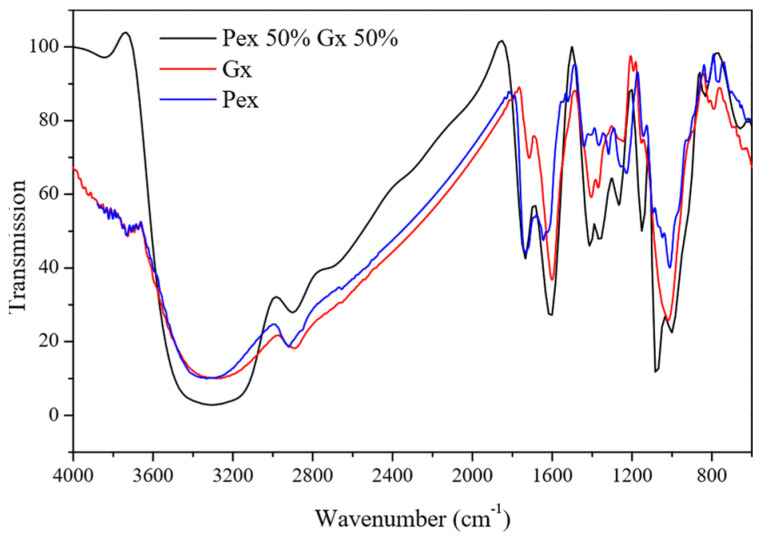
Infrared spectra of corrosion inhibitors: extracted pectin (Pex), xanthan gum (Gx), and the Pex 50%–Gx 50% mixture.

**Figure 5 molecules-27-02111-f005:**
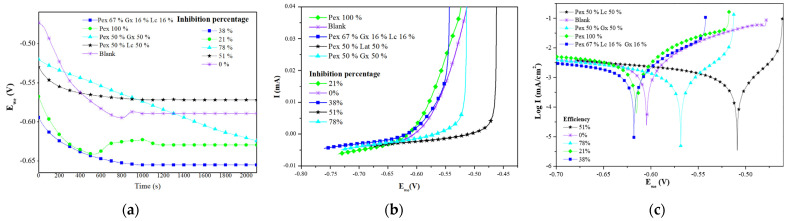
Curves from the linear polarization test of the formulations with the most prominent effect: (**a**) open circuit potential curves; (**b**) linear polarization curves; (**c**) Tafel curves.

**Figure 6 molecules-27-02111-f006:**
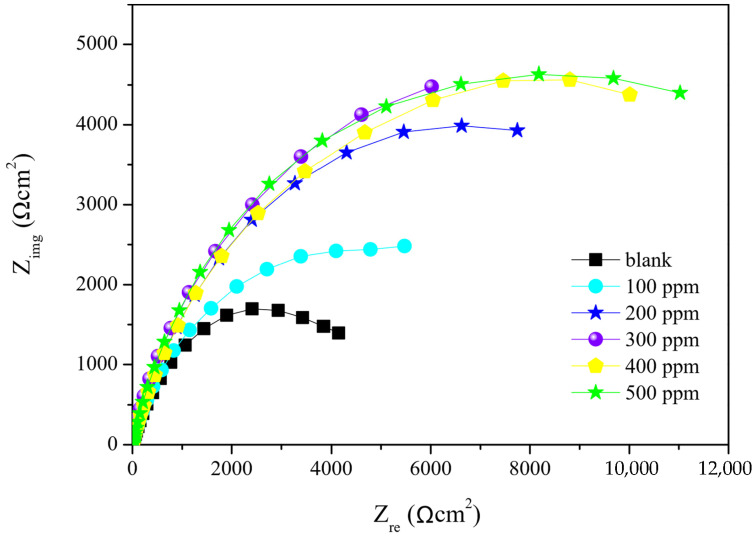
Nyquist impedance curves of the Pex 50%–Gx 50% formulation applied to the medium in different concentrations.

**Figure 7 molecules-27-02111-f007:**
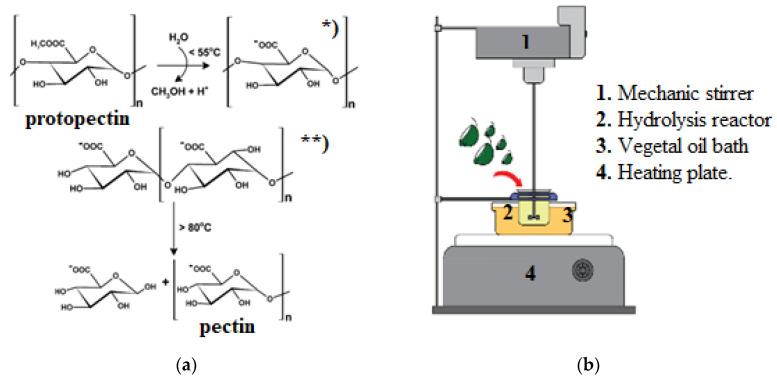
(**a**) Pectin hydrolysis mechanism: * protopectin demethylation, ** protopectin hydrolysis (Lim y Cheong 2015, modified by author); (**b**) pectin extraction system.

**Table 1 molecules-27-02111-t001:** Physicochemical characterization of King mandarin and Tahiti lime pectin.

**King Mandarin Pectin**				
**pH**	**PEq**	**%Met**	**%AUA**	**%DE**
3	1780.90 ± 186.56	8.85 ± 0.28	63.73 ± 3.41	79.50 ± 1.78
2	948.47 ± 63.56	8.36 ± 0.08	66.25 ± 0.79	71.74 ± 1.55
1	488.23 ± 4.12	7.66 ± 0.03	79.48 ± 0.44	55.25 ± 0.32
**Tahiti Lime Pectin**				
**pH**	**PEq**	**%Met**	**%AUA**	**%DE**
3	3135.22 ± 186.71	9.00 ± 0.32	71.21 ± 6.53	89.64 ± 0.98
2	857.80 ± 124.13	7.83 ± 0.29	72.66 ± 3.51	66.72 ± 1.93
1	489.11 ± 5.99	7.79 ± 0.08	81.17 ± 0.65	54.97 ± 0.47

**Table 2 molecules-27-02111-t002:** Corrosion inhibitor efficiency: commercial pectin (Cm), extracted pectin (Pex), and the 50%:50% formulation of Pex and Gx.

Concentration	Pectin Cm		Pex 50% and Gx 50%		Pectin Pex	
	Icorr	Ef	Icorr	Ef	Icorr	Ef
ppm	mA/cm^2^	%	mA/cm^2^	%	mA/cm^2^	%
Blank	32.26	0.00	32.26	0.00	32.26	0.00
100	8.31	75.14	15.06	41.10	20.98	35.06
200	7.52	76.32	14.46	53.23	14.34	57.16
300	6.53	83.31	7.22	77.18	17.23	45.03
400	7.64	75.27	6.60	78.21	22.35	29.20
500	9.14	71.14	7.71	73.25	25.53	26.14
600	13.24	57.08	12.84	65.11	25.98	25.17
700	13.07	59.27	16.46	47.21	25.36	26.36
800	14.66	54.23	16.54	47.36	25.54	26.12
900	13.87	59.18	16.91	46.35	21.51	27.14
1000	15.11	52.40	19.36	36.20	25.02	26.25

**Table 3 molecules-27-02111-t003:** Corrosion inhibition efficiency results of the Pex 50%–Gx 50% formulation.

Technique	LPR				EIS			Weight Loss	
Concentration	Potential Corrosion (Ecorr)	Current Intensity (Icorr)	Efficiency (Ef)	Charge Transfer Resistance (R_ct_)	Capacitance Double Layer (C_dl_)	n	Efficiency (Ef)	Corrosion rate	Efficiency Ef
(ppm)	(mV)	$\left(mA/cm^{2}\right)$	(%)	$\left(\Omega cm^{2}\right)$	$\left(\mu F/cm^{2}\right)$		(%)	(mmpy)	(%)
0	−649.94	32.26	0.00	502.67	2210.79	0.76	0.00	0.085	0.00
100	−564.79	15.06	41.10	826.38	1604.94	0.75	39.24	0.052	39.42
200	596.43	14.46	53.23	1087.77	1463.13	0.72	54.12	0.033	61.21
300	−665.84	7.22	77.18	1331.98	1194.87	0.79	62.23	0.026	69.06
400	−533.98	6.60	78.21	1830.22	579.72	0.70	73.15	0.015	82.15
500	−587.24	7.71	73.25	1646.5	644.41	0.72	69.58	0.016	81.23

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
