# Peer review of "Evaluation of Adding Natural Gum to Pectin Extracted from Ecuadorian Citrus Peels as an Eco-Friendly Corrosion Inhibitor for Carbon Steel"

_molecules, 2022, doi:10.3390/molecules27072111_

Round 1

Author Response

  1. The pictures in Figure 1 are not on an acceptable resolution for publishing. Please correct.

Figure 1 resolution has been updated.

  1. In Figure 1 there are no % for Lc for F7, F8 why?

According to the mixture experimental design, F7 and F8 resulted as binary formulation therefore Lc is not part of the mixture. Mixture design is based on the combinations given in Figure 1 in which there is a possibility to analyze just 2 from the 3 components.

Figure A. Mixture design scheme.( see attached file )

  1. Where is the explanation of the several steep peaks of the curve Cm in the DSC?

Explanation of the several peaks of the curve Cm is shown between lines 180 to 200 in the updated document, “DSC thermic profile of pectins Cm, LT and MK (Figure 2b) presents an endothermic peak between 120 and 128 °C that represents samples dehydration. Cm profile presents two endothermic steep peaks at 186 and 208 °C which are related to the breaking bond process of functional groups and result in polysaccharide chain depolymerization, which is also evident in its TGA profile where a weight loss of 45 % between 200 to 250 °C was observed [27].”

Reviewer 2 Report

Núñez-Morales et al. submitted a manuscript entitled “Influence of Natural Gums Addition to Pectin Extracted from Ecuadorian Citrus Peels in their Performance as an Eco-Friendly Corrosion Inhibitor of Carbon Steel in Saline Media” with a manuscript reference number: molecules-1425683 for the possible publication in the molecules. The authors described a particular class of corrosion inhibitors extracted from Ecuadorian Citrus Peels and their synergistic effects in the presence of xanthan gum as a corrosion inhibitor of carbon steel in saline media. There are many points that need to be clarified/included to a greater extent before the manuscript could be considered for publication in the molecules.

The specific comments are as follow:

  1. The title of the manuscript is not clear and too big; authors can modify the title in a more precise and shorter form.
  2. Page 2, line 48, authors used MW 20000 without any unit. The unit should be used.
  3. Page 3, line 82, why authors mentioned it is a proposal! These sentences should be modified.
  4. The novelty of the work is not clear. The authors mentioned that all these components could be used as a corrosion inhibitor then what advancement has been considered in this work. The novelty of the work should be clearly stated.
  5. The authors showed the individual efficacy of these extracted components. From the performance, it was decided to use Pex and Gx. However, it is not clear what they choose to use 50: 50 of these two. The authors should examine and discuss the variable ratio of these two and justify using 50:50 of Pex: Gx.
  6. Page 7, line 233: it was observed that the system without a corrosion inhibitor takes the shortest time to be stabilized (800 s). but figure 5a does not say that. A similar tendency can be found in others except for pex:lc. Therefore, the authors should rerun the EoC open circuit potential curves, see the stabilized region, and report the stabilization time.
  7. It is not clear from the manuscript what is the delay time was used for the electrochemical studies. It should be pointed out the delay time of collecting the electrochemical data.
  8. Page 7, line 239, what is sing sign of more. The sentence should be corrected.
  9. The authors reported inhibition efficiency by round number, but all other parameters with a decimal number. The significant figures should be maintained throughout the manuscript (e.g., Tables 2, 3).
  10. In the text, page 312, the IE has been reported as 72,53%. It is wrong. It should be fixed, and significant figures with similar decimal points should be used throughout the manuscript.
  11. Check the English with grammatical errors and types very carefully. For example, lines 318-320. Besides what it means “using de formulation.”
  12. Figure 1 is very vague. The figure should be self-explanatory. Figure 1 should be revised.
  13. Line 394: what is the meaning of 0,5 g! correct it!
  14. In the material and methods section, there should be a separate weight loss study. Currently, it goes under EIS, which is not correct! It should be separated.
  15. Nowhere is it mentioned what the composition of the carbon steel is?
  16. Notably, the report does not mention anything about the surface morphology of the steel in the absence and presence of inhibitors. The authors should study and include the surface morphology of the steel in the absence and presence of inhibitors.

Author Response

  1. The title of the manuscript is not clear and too big; authors can modify the title in a more precise and shorter form.

Title has been changed to “Evaluating the Addition of Natural Gum to Pectin Extracted from Citrus Peels as an Eco-Friendly Corrosion Inhibitor for Carbon Steel.

  1. Page 2, line 48, authors used MW 20000 without any unit. The unit should be used.

The cited paper does not mention the units of the molecular weight, however the cite has been changed and the units of the molecular weight of pectin were added.

  1. Page 3, line 82, why authors mentioned it is a proposal! These sentences should be modified.

Sentence has been removed.

  1. The novelty of the work is not clear. The authors mentioned that all these components could be used as a corrosion inhibitor then what advancement has been considered in this work. The novelty of the work should be clearly stated.

The novelty has been updated between lines 79-82 according to the comments “Based on these statements, the inhibitory efficiency of pectin extracted from citrus peels could be enhanced by adding natural gums (xanthan gums and latex from lechero plants), this mixture may result in an environmentally friendly corrosion inhibitor with higher protective performance compared to a synthetic conventional product.”

  1. The authors showed the individual efficacy of these extracted components. From the performance, it was decided to use Pex and Gx. However, it is not clear what they choose to use 50: 50 of these two. The authors should examine and discuss the variable ratio of these two and justify using 50:50 of Pex: Gx.

The ratio 50:50 of Pex and Gx is a result of the simple central three component mixture design applied to this study, which is explained in the Methodology item 3.5 lines 394-401. Besides, all the mixtures variables are shown in Figure 3. Furthermore, the ratio 50: 50 of Pex and Gx was selected for the next additional analysis (EIS, weight loss method) because it has shown the highest corrosion inhibition efficiency from all the other formulations in the mixture experimental design. Which is explained in the item. 2.3, lines 183-218.

  1. Page 7, line 233: it was observed that the system without a corrosion inhibitor takes the shortest time to be stabilized (800 s). but figure 5a does not say that. A similar tendency can be found in others except for pex:lc. Therefore, the authors should rerun the EoC open circuit potential curves, see the stabilized region, and report the stabilization time.

Figure 5a was modified to show the stabilization time clearly, in which is evidenced that the system without a corrosion inhibitor takes de shortest time to be stabilized, and that the system with the corrosion inhibitor with the highest efficiency shows the longest stabilization time.  

Figure 5a.   Eoc open circuit potential curves.(see figure in the attached file)

  1. It is not clear from the manuscript what is the delay time was used for the electrochemical studies. It should be pointed out the delay time of collecting the electrochemical data.

Section 3. Materials and methods. Item 3.5.1 LPR electrochemical evaluation. "Carbon steel oxidation potential range was determined by cyclic voltammetry (VC) and resulted in ±110 mV around the steady-state open circuit potential (EoC). The corrosion system was stabilized for 20 min and the LPR analysis was performed immediately for each formulation given with a potential rate of 1 mV/s. Tafel slopes were obtained as test results and the inhibition efficiency (η%) was calculated according to Equation (5)"

In this paragraph is shown that the system was stabilized at 20 min, which is the delay time considered for the corrosion cell to allow the mass transfer and adsorption of the inhibitor on the metal surface.

  1. Page 7, line 239, what is sing sign of more. The sentence should be corrected.

It has been updated to “the equilibrium is a sing of more stable corrosion products formation; that is to say, the curves belonging to the blank and Pex 100% ".

  1. The authors reported inhibition efficiency by round number, but all other parameters with a decimal number. The significant figures should be maintained throughout the manuscript (e.g., Tables 2, 3).

All the parameters were unified to two decimal numbers.

  1. In the text, page 312, the IE has been reported as 72,53%. It is wrong. It should be fixed, and significant figures with similar decimal points should be used throughout the manuscript.

The observation was revised and corrected according to Table 3. “The highest value of corrosion inhibition was obtained by adding the formulation of 400 ppm resulting in an efficiency of   73.15% (Table3)".

  1. Check the English with grammatical errors and types very carefully. For example, lines 318-320. Besides what it means “using de formulation.”

The manuscript has been thoroughly revised by a native English speaker and professional academic editor.

  1. Figure 1 is very vague. The figure should be self-explanatory. Figure 1 should be revised.

Figure 1 has been modified.

  1. Line 394: what is the meaning of 0,5 g! correct it!

The number has been updated

  1. In the material and methods section, there should be a separate weight loss study. Currently, it goes under EIS, which is not correct! It should be separated.

Item 3.5.3. Weight loss method has been added to separate this procedure from Item 3.5.2. Electrochemical impedance spectroscopy.

  1. Nowhere is it mentioned what the composition of the carbon steel is?

The composition of the carbon steel was included in Appendix B, lines 472 -476

  1. Notably, the report does not mention anything about the surface morphology of the steel in the absence and presence of inhibitors. The authors should study and include the surface morphology of the steel in the absence and presence of inhibitors.

This study was completed in 2020 as part of a research project, the morphological study of the surface had been planned to be performed in a complementary study to the electrochemical analyzes. However, due to the pandemic, they have not yet been carried out, therefore, this study will be submitted in a future publication. Despite it, the electrochemical analyzes allow the analysis in-depth of the corrosion inhibition effectiveness.

Reviewer 3 Report

In this study, the influence of natural gums addition to pectin extracted from ecuadorian citrus peels in their performance as an corrosion inhibitor of steel in saline media. It is a topic of interest to the researchers in the related area. Whereas the linguistic writing method is in urgent need of careful review. The following major corrections are needed before possible publication of the submitted research manuscript:

  1.     The English in this paper needs to be greatly improved.
  2.   The introduction part is poorly written and needs to be greatly improved to show the innovation of the manuscript. The recent and professional references should be cited “Corrosion Science 133 (2018) 6–16, Corrosion Science 191 (2021) 109715, Journal of Materials Science & Technology 52 (2020) 63–71”
  3. All the graph is not clear. 
  4.   The graph number is chaotic. In page 11, is it Fig. 1 ??? 
  5.    In the Fig. 5, (a) is the same as (b), what could make a writer make such mistakes? 
  6. The PDP and EIS curves for three inhibitors at different concentrations should be provided. It’s is important. 
  7.   Particularly, the detail and in-depth analysis should be given in PDP and EIS section.
  8.  The conclusion part should be rewritten to make it fluent.

Author Response

Thank you for your comments in the following you will see the corresponding answers.

  1. The English in this paper needs to be greatly improved.

The manuscript has been revised by a native English speaker and a professional academic editor; the certificate of English editing is included.

  1. The introduction part is poorly written and needs to be greatly improved to show the innovation of the manuscript. The recent and professional references should be cited “Corrosion Science 133 (2018) 6–16, Corrosion Science 191 (2021) 109715, Journal of Materials Science & Technology 52 (2020) 63–71”

Corrosion Science 133 (2018) 6–16 was included in the introductory part to improve the versatility and effects of plant extracts for corrosion inhibition of metals.

  1. All the graph is not clear. 

Figure 1, Figure 5a and 5b were improved.

  1. The graph number is chaotic. In page 11, is it Fig. 1 ??? 

The numbering of the figures has been updated.

  1.  In the Fig. 5, (a) is the same as (b), what could make a writer make such mistakes? 

  Figure has been replaced by the corresponding graph.

  1. The PDP and EIS curves for three inhibitors at different concentrations should be provided. It’s is important. 

Lineal polarization results LPR are shown in Table 2. In this study, impedance spectroscopy and weight loss analysis were performed on the samples with the highest corrosion inhibiting effect at different concentrations (Table 3) as a complement to the first part of the study, which was the determination of the sample with the highest inhibitory effect. In order to be practical, it was considered to carry out the complementary studies (EIS and weight loss) only to the samples with the greatest inhibitory effect since the other samples of the mixture experimental design were discarded from the rest of the global experimental design.

  1. Particularly, the detail and in-depth analysis should be given in PDP and EIS section.

The discussion presented was carried out based on the results obtained from the EIS analysis in order to remark the presence of adsorption processes and changes on the electronic transfer of the system by the corrosion inhibitor performance.

  1. The conclusion part should be rewritten to make it fluent.

The conclusions have been updated.
